# Evaluation of the Gelation Characteristics and Printability of Edible Filamentous Fungi Flours and Protein Extracts

**DOI:** 10.3390/foods14060923

**Published:** 2025-03-08

**Authors:** Lauren Doyle, Suvro Talukdar, Youling L. Xiong, Akinbode Adedeji, Tyler J. Barzee

**Affiliations:** 1Department of Biosystems and Agricultural Engineering, University of Kentucky, Lexington, KY 40546, USA; laurenedoyle@uky.edu (L.D.); suvro.talukdar@uky.edu (S.T.); akinbode.adedeji@uky.edu (A.A.); 2Department of Animal and Food Sciences, University of Kentucky, Lexington, KY 40546, USA; youling.xiong@uky.edu

**Keywords:** mycoprotein, hydrogel, 3D printing, future foods, protein extraction

## Abstract

There is a pressing need to produce novel food ingredients from sustainable sources to support a growing population. Filamentous fungi can be readily cultivated from low-cost agricultural byproducts to produce functional proteins for food biomanufacturing of structured products. However, there is a lack of scientific knowledge on the gelling characteristics of fungal proteins or their potential in additive biomanufacturing. Therefore, this study investigated the feasibility of utilizing fungal protein extracts and flours from *Aspergillus awamori*, *Pleurotus ostreatus*, *Auricularia auricula-judae* as sole gelling agents in 3D-printed products. Protein extracts were successfully prepared using the alkaline extraction–isoelectric precipitation method and successful physical gels were created after heating and cooling. Results indicated that shear-thinning gel materials could be formed with acceptable printability at mass inclusion rates between 15% and 25% with the best performance obtained with *P. ostreatus* protein extract at 25% inclusion. *A. auricula-judae* demonstrated promising rheological characteristics but further optimization is needed to create homogeneous products appropriate for extrusion-based 3D printing. This work provides valuable insights for continued development of 3D-printed foods with filamentous fungi.

## 1. Introduction

As the global population continues to grow, the demand for protein also continues to rise [1]. Technological advances have enabled the production of highly customizable and versatile materials with applications in many areas of sustainable food systems. Because of this, knowledge of the functional properties and characteristics of food ingredients has become increasingly important. Finding innovative ingredients and/or processes (e.g., 3D bioprinting) to improve the functional properties (e.g., emulsification, foaming, gelation, water- or oil-holding capacity) of foods is essential. A promising potential source of sustainable functional food ingredients are microorganisms such as filamentous fungi. Fungi represent potentially low-cost and eco-friendly materials that grow rapidly and may be cultivated on diverse substrates including organic byproducts of agriculture and food processing industries [2]. Many fungi possess relatively high protein contents of up to 45% of the total biomass [3] and fungal biomass fractions have been found to possess biochemical compounds with high functional properties [4,5]. For instance, centrate co-products of the mycoprotein Quorn^TM^ fermentation process (using filamentous fungus *Fusarium venenatum*) containing mycelial aggregates, proteins, phospholipids, chitin, nucleotides, and other cell membrane and cell wall constituents were found to display useful rheological, emulsifying, and foaming properties [4,6].

Fungi have also been used as bioink mixtures to add nutritional and functional characteristics to 3D-bioprinted constructs. Three-dimensional bioprinting is an emerging technique in food manufacturing that creates precise 3D shapes with customizable dimension/shape, flavor, texture, and nutritional profiles [7,8,9]. The technology could be capable of providing personalized nutrition, delivering sensitive/easily degradable bioactive compounds, modifying food textures (e.g., for individuals suffering from dysphagia), printing 3D foods in remote locations (e.g., astronauts or soldiers) [7,9,10,11]. Three-dimensional bioprinting also has significant potential in the creation of cultivated meat and other pure or hybrid alternative protein products [10,12,13,14]. Filamentous fungi have been utilized in 3D food bioprinting applications for a variety of functions making use of their unique morphological structure and composition. Wang and Liu [15] used an addition of 4.18% shiitake mushroom powder to printable bioinks which comprised wheat gluten (10.45%), Tween-80 (1.39%), water (65.00%), cocoa butter (2.32%), starch from potato or wheat (4.64%), protein isolate from soy or pea (10.45%), and sodium alginate (1.57%). In another study, black fungus (*Auricularia auricula*) was used to create gels for dysphagia patients utilizing a water-based solution of black fungus powder with additions of arabic gum, xanthan gum, and k-carrageenan gum as stabilizer components [9]. Santhapur et al. [16] utilized fungal mycoprotein (mycelium) and potato protein to create hybrid food products that made use of the inherent fibrous nature of the mycelium (mimicking muscle fibers) and the physical gelling nature of the potato protein to create meat-like structures and textures. Living fungi have also been used to create self-healing packaging materials with bioinks composed of agricultural feedstock inoculated with mold-type fungus, which after initial colonization, was reduced in particle size and added at 20% to a mixture of water and 4% psyllium husk powder, printed, and observed for re-colonization behavior [17].

Hydrogels are important materials for foods and the formation of a 3D structure or scaffold is useful for bioprinting applications. Gelatin and sodium-alginate are popular hydrogel materials [18] and where sodium-alginate is principally derived from brown algae, commercial gelatin is almost exclusively produced from animal-derived sources, which limits its application in alternative protein products [19,20]. Fungal-derived chitin and chitosan have been used in the creation of hydrogels [21], but the process of extracting chitin followed by the deacetylation to chitosan can be arduous, and the resulting material may lack adhesion factors necessary for the attachment of animal cells in cultured meat applications [22]. On the other hand, fungal protein extracts exhibit native attachment factors for animal cells [23]. Protein extracts from filamentous fungi are a compelling fraction to explore for functional properties since protein extracts from plants and animals have found use in many food products due to their emulsification, foam formation, water-holding capacity, oil absorption, and gelation properties [24]. These alternative protein isolates include insects [24], soy protein [25], and pea protein [26], among others. However, the utilization of fungal protein extracts for the development of novel food products is underexplored. Edible fungal species protein contents can range from 9 to 52% protein, compared to that of grain legumes of 18–34%, cereal grains and pseudocereals of 8–11%, oilseed by-products (e.g., press cakes or solvent-extracted flours) of 45–65%, and green biomass (leaves, seaweed, microalgae, aquatic plants) of 12–70% [3]. Green biomass, while high in protein content, contains mainly structural and biologically active proteins [27] with a low shelf-life [28], low extraction yields/purity [3,29], a presence of off-flavors [29], and lower digestibility [30]. Fungi can be grown on agricultural by-products with high efficiency, are commonly used in foods, and can display high Protein Digestibility-Corrected Amino Acid Scores (PDCAAS; e.g., mycoprotein from *F. venenatum* of 0.91) [3,31,32]. Certain fungal strains have also demonstrated high globulin contents, such as *Pleurotus ostreatus*, with 62% of its total protein content (24.7%), which indicates promise for use as a gelling agent [33]. *Auricularia auricula-judae* has water soluble protein—polysaccharide that can act as a food thickener by improving texture and modifying starch gelatinization and gluten structure [34].

Filamentous fungi represent promising functional materials for future foods. However, there is currently a gap in knowledge on the functional characteristics of fungal extracts and the procedures required for applying such materials in additive biomanufacturing and cellular agriculture applications. Therefore, this study’s objectives were to (1) develop processes to create functional fungal protein extracts from different fungal strains, *Aspergillus awamori*, *Pleurotus ostreatus*, *Auricularia auricula-judae*, and evaluate their characteristics, and (2) quantify the printability of the resulting fungal gels for the creation of structured food precursor materials.

## 2. Materials and Methods

### 2.1. Materials

Dried mushroom samples of *Pleurotus ostreatus* and *Auricularia auricula-judae* were obtained from Mushroom House (Westbury, NY, USA). The biomass was stored at room temperature in an air-tight container prior to use. *Aspergillus awamori* (ATCC 22342) was cultivated according to published methods for spore cultivation and inoculation [35]. Briefly, spores were aseptically inoculated to 24 g/L of potato dextrose broth (PDB) (Becton, Dickinson and Company, Sparks, MD, USA) medium for submerged culture in a New Brunswick Bioflo 10 L fermenter (Eppendorf, Hamburg, Germany) with an inoculum of 10^3^ spores/mL. The fermenter parameters were set to the temperature of 30 °C, which was controlled with a water jacket heating/cooling unit, with a stir speed of 200 rpm by a Rushton turbine. The system was run for 72 h and the biomass was harvested by straining and rinsing with deionized water prior to storage at −40 °C until further use. All chemical reagents used were of analytical grade.

### 2.2. Flour Processing and Protein Extraction

The *A. awamori* was lyophilized using a LabConco FreeZone 4.5 L −105 °C Model 72040 (Kansas City, KS, USA) for 72 h while other fungi samples were obtained in dried form and thus used as received. Dry fungi samples were processed using a Vitamix A3500 blender (Olmsted Township, OH, USA) on speed 5 for 60 s, resting for 3 min, then blended on speed 10 for 30 s to disrupt the cell walls and increase the bioavailability of functional materials in the fungi. The contents were then filtered through a 40-mesh sieve (0.4 mm) and the resulting flour was collected in an air-tight container and stored at −40 °C until use. Bulk density was measured using standard methods by weighing the dried fungal materials and recording their respective volume, then adding an additional weighed powder and recording the new volume, which was repeated three times for each material [36]. Once the final volume was reached, the material was tapped by physical means until the powder no longer compacted.

An alkaline extraction–isoelectric precipitation procedure was used to generate protein extracts from fungal flours (Figure A1). A 5:1 ratio of deionized water and fungal flour was mixed at high speed in the Vitamix A3500 blender for 30 s to homogenize, reduce in particle size, and further disassociate the cell walls. The pH was then increased to 11.5 by adding 1 M NaOH to the slurry and then blending again on the medium (5) speed setting for 15 min. The slurry was then centrifuged at 4250× *g* for 10 min and the supernatant was collected. The pH of the supernatant was decreased to the isoelectric point of 4.5 using 1 M HCl and the mixture was transferred to an Erlenmeyer flask and gently agitated at 100 rpm for 30 min. The mixture was again centrifuged at 4250× *g* for 15 min, the supernatant was discarded, and the pellet was neutralized using 1 M NaOH. The neutralized pellet was then lyophilized for 24 h and stored at −40 °C until further use. The pellet collected from the initial centrifugation step was diluted at 2:1 with deionized water and reprocessed through the alkaline extraction–isoelectric point treatment to increase the extraction yield. This step was repeated three times.

Throughout this article the following material abbreviations are used: AF (*A. awamori* flour), AP (*A. awamori* protein extract), OF (*P. ostreatus*, oyster mushroom, flour), OP (*P. ostreatus*, oyster mushroom, protein extract), WF (*A. auricula-judae*, wood ear mushroom, flour).

### 2.3. Material Characterization

#### 2.3.1. Protein Content

Carbon, nitrogen, and protein contents of fungal flour and protein extract samples were determined by the Dumas combustion method (FlashEA 1112, Thermo Fisher Scientific Inc., Waltham, MA, USA). The nitrogen content was converted to protein content by a conversion factor of 6.25 [5]. Protein yield was measured as the lyophilized mass (g) of protein extract obtained divided by the mass (g) of original dry biomass. Protein recovery was calculated as the protein content in the extract divided by the protein content in the original fungal flour. Protein purity was measured as the protein content in the protein extract.

#### 2.3.2. SDS-PAGE

Sodium dodecyl sulfate–polyacrylamide gel electrophoresis (SDS-PAGE) was run using Mini-PROTEAN TGX precast gradient gels (4–15% acrylamide), 10 well, 50 μL/well (Bio-Rad, Hercules, CA, USA) to evaluate the specific protein size within the fungal protein extract. The *A. awamori* and *P. ostreatus* protein extracts were dissolved in deionized water to a final protein content of 3 mg/mL. Concentrated (4x Laemmli protein sample buffer (Bio-Rad Cat. #1610747, Hercules, CA, USA) was added in a 1:4 ratio with the dilute protein extract solution. Half the samples had 5% *v/v* of the reducing agent (RA) β-mercaptoethanol added to the sample buffer [37]. An enzymatic protease inhibitor cocktail (EDTA-free 100x) (Halt, Pierce Biotechnology, Rockford, IL, USA) was added at 10 μL/mL to half of the samples [37,38]. Bovine gelatin of concentration 3 mg/mL was tested as a positive control, with only the sample buffer added. All samples were then boiled for 3 min for complete protein dissociation and SDS binding. Each well contained 20 μL of material. The gels were run at 90 V for 15 min to initialize movement, and 110 V for 60 min until the stain line was 5 mm away from the bottom of the gel. After electrophoresis, the gels were stained with 1x Bio-Safe Coomassie blue protein stain (Bio-Rad, Hercules, CA, USA) and shaken at 50 rpm for 1 h. The gels were destained using DI water and shaking at 50 rpm for 48 h, with water replacement every 12 h.

#### 2.3.3. ATR-FTIR

The dried protein extracts and flours were analyzed using an Attenuated Total Reflectance-Fourier Transform Infrared (ATR-FTIR) spectrometer (Nicolet Is50, Thermo Fisher Scientific Inc., Waltham, MA, USA) equipped with a deuterated L-alanine doped triglycine sulfate (DLaTGS) detector. The collected spectra were in the range of 500–4000 cm^−1^, using a resolution of 4 cm for a total of 32 scans averaged for each sample. OMNIC software (Version: 9.13, Nicolet Instrument Corporation, Waltham, MA, USA) and GraphPad Prism 10 (San Diego, CA, USA) were utilized to analyze the spectra and to remove noise, respectively. Noise reduction was performed utilizing 2nd order polynomial with four adjacent-sample-averaging in GraphPad Prism. The samples were run in triplicate. An interpretation of the spectral peaks was performed in correspondence with that of Movasaghi et al. [39].

#### 2.3.4. Particle Size and Zeta Potential Analysis

The particle size and zeta-potential were measured with a Malvern Panalytic Zetasizer Nano ZS (Westborough, MA, USA) in dynamic light scattering mode with a procedure adapted from Hunter et al. [40]. Prior to analysis, the dried fungal materials were diluted to 0.05 mg/mL in water and adjusted to a pH of 7 using 0.1 M NaOH and vortexed before each reading. The data were fitted to a Gaussian distribution model in Graphpad Prism to be statistically evaluated for mean and standard deviation.

#### 2.3.5. Color

A color analysis of hydrated flour and protein extract samples was performed using ImageJ^TM^ software (version: 2.16.0) to quantify the color parameters (L*, a*, b*) from digital images obtained under consistent lighting conditions [41]. Images were first converted from the RGB color space to the LAB color space using the “Color Space Converter” plugin in ImageJ. The resulting stack was split into three grayscale images corresponding to the L*, a*, and b* channels (lightness/darkness, red/green, and yellow/blue, respectively). Mean pixel intensity values for each channel were obtained by analyzing selected regions of interest.

### 2.4. Minimum Gelation Concentration

The hydrogels were prepared by adding 15%, 20%, and 25% (g fungi/g total wet basis) of each flour or protein extract to 5 g deionized water in a 15 mL centrifuge tube. Each solution was then homogenized with a Vitamix A3500 blender. The gelation process was performed according to the methods of Ma et al. [42], with sealed tubes placed in a water bath and heated at a rate of 1 °C/min to a final temperature of 70 °C, based on the denaturation point of *S. cerevisiae* proteins [43]. The heating was performed using a temperature feedback module water bath. The contents of each tube were then placed into sealed syringes, covered, and chilled in an ice bath placed in the refrigerator (4 °C) for 24 h to set fully. To determine the minimum gelation concentration, similar methods were followed [44,45]. Briefly, heated and cooled samples were inverted in their tube and observed for material that slid down or stayed in place as “failed” or “passing” gels, respectively. The gels were allowed to equilibrate to room temperature before further analysis.

### 2.5. Rheology

Rheological properties of fungal flours and protein extracts were assessed with a Discovery hybrid rheometer (DHR-2, TA Instruments, New Castle, DE, USA) with a 40 mm plate geometry. Viscosity was determined over a shear rate range of 1–100 s^−1^ at a constant temperature of 25 °C. The samples were prepared in 15% and 25% inclusion rates following the same procedures as the Minimum Gelation Concentration experiment. Apparent viscosity was modeled with the power law model(1)$$\eta =K{\dot{\gamma }}^{\left(n-1\right)}$$where η is the apparent viscosity (Pa∙s), K is the consistency coefficient (Pa∙s^n^), $\dot{\gamma }$ is the shear rate (s^−1^), and n is the flow index [16,46,47]. Data were analyzed with least squares regression with the GraphPad Prism 10 (San Diego, CA, USA) software. Data in the ‘Newtonian plateau’ (when shear stress is significantly low) were excluded as expected. The plot of the log of apparent viscosity against the log of shear rate allowed the determination of the power law parameters (K and n).

### 2.6. Three-Dimensional Bioprinting

Fungal hydrogel materials were 3D-printed using an Allevi3 Bioprinter (Philadelphia, PA, USA) to demonstrate a potential application in the creation of structured constructs. The pressure and printing speed was determined both subjectively and objectively. The pressure was determined by adjusting the pressure and observing when the extrusion was smooth and flowing consistently from the needle (Table 1). The optimal printing speed was determined by extruding the gel at the ideal pressure for 3 s and measuring the length of the extruded bit. Printing patterns were adapted from Zhang et al. [48] using a six-pointed star of 10.2 mm width and 3.8 mm overall height with a layer height of 0.7 mm. The fungal hydrogels were printed in duplicate.

### 2.7. Texture Profile Analysis

Texture profile analysis (TPA) was carried out using a TA XT plus Texture Analyzer (Texture Technologies Corp., Hamilton, MA, USA). The analysis was conducted at room temperature (25 °C). The method for the TPA used was adapted from Xu et al. [49]. The analysis was performed on the successfully 3D-printed samples immediately after printing. A double compression cycle was applied using an acrylic cylindrical probe measuring 1 mm in diameter. The protocol included a 5 g trigger speed, 1 mm indent (26% compression), and 1 mm/s pre-, post-, and during-test speed. Force–time deformation curves were collected and evaluated to determine hardness, cohesiveness, gumminess, and resilience using the raw collected data [50].

## 3. Results and Discussion

### 3.1. Flour Processing, Protein Extraction, and Color

In the fungal protein extraction process, the primary objective was to devise a straightforward and scalable methodology. The utilization of an alkaline extraction–isoelectric precipitation technique offers notable advantages, as it uses only HCl and NaOH, both of which become inert post-process, yielding NaCl and H_2_O. The procedural steps are uncomplicated and readily scalable to industrial scales. Throughout the protein extraction procedure, the fungal flours underwent solubilization and precipitation, as well as a pH-dependent color change (Figure A2). The overall yield (g protein extract/g dry biomass loaded) was 0.16 g/g, 0.09 g/g, and negligible for *A. awamori*, *P. ostreatus,* and *A. auricula-judae*, respectively (Table 1). Due to the high polysaccharide content of *A. auricula-judae*, the solution became extremely viscous and led to poor separation of fungal flour even under high centrifugation forces (12,500× *g*). This lack of separation did not allow for the successful protein extraction for *A. auricula-judae*. The protein content of the initial *A. auricula-judae* flour was very low at 6.0% compared *to A. awamori* and *P. ostreatus* flour protein contents of 29.3% and 26.0%, respectively. Due to the low initial protein content and poor water-soluble protein extraction performance, *A. auricula-judae* protein extract was not explored further.

Overall, the total protein recovery (protein content in the extract divided by the protein content in the fungal flour) was 32.2% and 18.4% for *A. awamori* and *P. ostreatus*, respectively. When comparing the protein concentration in the fungal flours and extracts, the extraction process increased the dry basis protein concentration by approximately two times—from 29.3% to 59.0% and from 26.0% to 51.5% for *A. awamori* and *P. ostreatus*, respectively. The purities of the protein extracts were low compared to plant extracts such as soybean protein isolate (91.4%) [51], and pea protein isolate (87.3%) [52] obtained with similar isoelectric precipitation extraction methods. Besides protein, the fungal extracts likely contained other cellular materials such as polysaccharides and pigments.

The low protein recoveries obtained in the present study can be explained by challenges in lysing fungal cell walls to expose intracellular proteins. The chitinous cell walls of fungi are meant to penetrate materials, requiring significant hardness to do so [53]. The means to extract protein by thoroughly degrading cell walls have been explored in the literature. For instance, breaking down cell walls using mechanical means like ultrasonication and high-pressure homogenization [5], or chemical processes like the addition of strong chemical buffers or enzymatic digestion [54,55] have the ability to increase protein recovery. Krishnaswamy, Barnes, Lotlikar and Damare [55] investigated different extraction buffers and lysing techniques, with Tris-MgCl_2_ buffer and homogenization with zirconium beads. Another protein extraction method for filamentous fungi used lyophilization and manual grinding to lyse the cell walls before utilization of a Tris-glycine buffer solution to extract the protein with the expected results from 500 mg of *Metarhizium anisopliae* of 0.03–0.20 mg protein/mg biomass (15–100 mg/mL) [56]. Zeng, Nilsson, Teixeira and Bergenståhl [5] used high-speed homogenization at a pH of 10 to lyse cells along with other cell wall degradation techniques such as high-pressure homogenization, enzymatic degradation, and ultrasonication prior to isoeletric-point precipitation, yielding a final protein recovery of 77%. The impacts of such techniques on gelation behavior and 3D printing are topics for future study.

The flour and protein extracts differed in their bulk density with protein extracts displaying increased bulk density compared to their flour counterparts (Table 2). The *A. auricula-judae* flour displayed the highest bulk density of 0.90 g/mL while the *A. awamori* flour had lowest density, at 0.20 g/mL.

### 3.2. SDS-PAGE Protein Characterization

SDS-PAGE provides the molecular weight profile of the protein present in the crude fungal protein extract (Figure 1). Observing the position and thickness of the bands created by the electrophoresis process can show insight into the specific protein breakdown. β-mercaptoethanol reducing agent was added to wells #4 and #5 (*A. awamori*), and #8 and #9 (*P. ostreatus*) to observe changes related to disulfide bonds, an important aspect of protein gelation. Ma et al. [43] illustrated the effect of β-mercaptoethanol on disulfide bonds in pea protein isolate with the reduction in larger proteins compared to samples without β-mercaptoethanol. The same trend is seen when comparing between wells #2/3 and #4/5 (*A. awamori*) as well as #6/7 and #8/9 (*P. ostreatus*). The *P. ostreatus* shows a large proportion of the disulfide bonds that occur within the protein extracts in the range of 75–250 kDa. The comparison between wells #2/3 and #4/5 indicates that *A. awamori* protein extract disulfide bonds were minimal. Bands at 245 kDa are reduced to thicker bands at 75 kDa with the addition of β-mercaptoethanol indicating that the disulfide-linked proteins were primarily in that molecular weight region. Work by Effiong et al. [57] showed a presence of sulfur-containing amino acids, methionine and cysteine, for *P. ostreatus*. Comparison of the fungal protein extract SDS-PAGE assays in Figure 1 to that of other SDS-PAGE, found in the literature, shows there is a common theme of a wide molecular range. Okeudo-Cogan et al. [58] used the hyphal protein from *F. venenatum* and observed an even distribution of protein concentrated between 8 and 136 kDa, with a defined band at 70 kDa.

### 3.3. ATR-FTIR

FT-IR is a spectroscopy technique that quantifies how the chemical bonds in a material absorb infrared light and can be used to quantify the functional groups present in the material [39,59]. Figure 2 demonstrates conserved spectral peaks for all samples at 3300, 2900, 1650, and 1000 cm^−1^. The first peak at 3300 cm^−1^ was identified as either nitrogen–hydrogen (N–H) bonds or a hydroxyl (O–H) functional group. Common organic alcohols, carboxyl acids, and amines contain N–H and O–H bonds. The second peak at 2900 cm^−1^ was identified as N–H bonds and/or alkane functional groups (C–H). The compounds in this range are amine salt and alkanes. The third peak at 1605 cm^−1^ indicates amide I or II functional groups. This group is a component of protein. The fourth peak at 1000 cm^−1^ indicated (C–O). In this region, glycogen and other carbohydrate-related C–O stretches are found. The dietary fiber of mycoprotein flours make up approximately 15–25% of the dry biomass [31] which is further composed of approximately 33% chitin (*N*-acetylglucosamine) and 66% β-glucans (1,3-glucan and 1,6-glucan) [60]. Overall, the spectra obtained were consistent with our and other’s prior work with *Aspergillus awamori* and other fungi [35,59,61] and corresponded closely with protein extracts of *Agaricus bisporus* mushroom stems [62].

The *A. awamori* and *P. ostreatus* flours produced similar spectra, where the spectra of the two protein extracts were also similar. The flours were dominated by N–H, O–H, and C–O bonding (3300 and 1000 cm^−1^, respectively). Notably, the *P. ostreatus* protein extract spectra had the lowest overall transmittance intensity, and both protein extracts displayed lower transmittance intensities around 3300 and 1000 cm^−1^ (predominately N–H, O–H, and C–O bonds), which was hypothesized to indicate the removal of chitin (*N*-acetylglucosamine) during the protein extraction procedure. However, the protein extracts still displayed strong signals of C–O and O–H bonds, which could indicate the presence of residual chitin and β-glucans in protein extracts and corresponds to the relatively low protein purities obtained in this study. A prior study of alkali extracts of *P. ostreatus* observed polysaccharides of (1→3)-α-d-glucan and branched (1→3)(1→6)-β-d-glucan mixed with protein [63], which is consistent with the present study. Further purification of fungal polysaccharides was recommended to involve hot and/or cold water and dimethyl sulfoxide extraction steps. Similar compositions of α-1,3- and β-1,3-glucans and chitin were found for two *Aspergillus* species along with mobile galactosaminogalactan (GAG), galactomannan (GM) [64], the latter of which has also been noted in *P. ostreatus*. Along with protein interactions explored in the present work, it is likely that polysaccharide interactions also influence the material characteristics.

### 3.4. Zeta Potential and Particle Size of Flours and Protein Extracts

Zeta potential represents the effective surface charge of the particles and quantifies the tendency of particles to engage in electrostatic interactions and ultimately flocculate or repel each other. Particles of neutral zeta potential (0 mV) will favor van der Waals attractive forces over electrostatic repulsions and tend to flocculate and settle [65]. Figure 3 and Table 3 display the average particle size and zeta potential at a pH of 7 for 40-mesh (0.4 mm) sieved protein extract and flour samples. The mean particle diameter of protein extract samples was lower than the corresponding flours for both *A. awamori and P. ostreatus.* The zeta potential of all samples was found to be negative at pH 7, as expected. *A. auricula-judae* flour demonstrated a distinct bimodal zeta potential which, due to its known high polysaccharide content, may indicate a distinct portioning of particles between neutral polysaccharides and other biomass components. Overall, the highly negative zeta potential obtained was expected and similar to previous results of oyster and shiitake mushrooms [66] and other mycelium materials (mycoprotein and *Aspergillus awamori*) [16,61]. It should be noted that the zeta potential is pH dependent and prior work of fungal flours and potato protein has demonstrated the potential to create electrostatically attracted materials by empirically determining an effective pH value that creates oppositely charged materials [16]. In principle, this effect could enable different structures and textures of hybrid fungal products and is an interesting area for further work.

### 3.5. Hydrated Flour and Extract Color and Minimum Gelation Concentrations

Hydrated flour and protein extracts before heating can be seen in Figure 4. The primary qualitative observations included the texture and color of the materials. The texture of the *A. awamori* flour created a dough with the addition of 5 mL of water, while the *A. auricula-judae* flour appeared to have a grainy texture and thickness similar to the *A. awamori* flour. However, the texture of the *A. awamori* flour was smooth and well combined, while the *A. auricula-judae* dough was grainy and separated. The other three samples displayed similar textures at each inclusion rate with the exception of 20% *A. awamori* protein extract, which was thicker and less smooth. The color parameters (L*, a*, b*) of fungal flour and protein extract materials varied significantly based on the material’s source, type (flour or protein extract) and inclusion level (Table 4). For *Aspergillus awamori* and *Pleurotus ostreatus* samples, flour materials showed higher L* and b* values than their protein extract counterparts, suggesting that flour contributed to a lighter and more yellowish appearance in both cases. Across all samples, *A. awamori* flour exhibited the highest brightness, while *A. awamori* protein extract showed a marked decrease in L* with increasing inclusion levels, from 38.2 (15% inclusion) to 23.6 (25% inclusion), indicating significant darkening. *P. ostreatus* flour had the highest a* values, suggesting a stronger red hue, while *A. auricula-judae* flour exhibited the lowest b* values, indicating a lower yellow color intensity. The observed color difference between flour and protein extracts could involve fungal pigments like carotenoids, melanin, polyketides, and azaphilones [67,68].

The minimum gelation concentration (MGC) was observed by inverting the heated gels and noting if the materials adhered to the sides or slid down (Figure 4). The three treatments of *P. ostreatus* flour displayed of the possible situations with the 15% treatment completely falling to the bottom of the inverted syringe (F indication), 20% partially sliding down the side (~ indication), and 25% adhering to the top of the syringe with no indication of sliding (P indication). The *A. auricula-judae* flour texture remained similar to the pre-heated mixture, maintaining the separated graininess. The presence of this texture suggested the existence of large particles in the mixture, which was confirmed from particle size analysis and resulted in a lack of homogeneity in these samples and a risk of clogging the printing needle. The MGC of *A. awamori* protein extract and flour were 20% and 15%, respectively. *P. ostreatus* protein extract was comparable to that of *A. awamori* protein extract at an MCG of 20%, but the flour performed much worse with an MCG of 25%. When comparing to the literature, Cruz-Solorio, Villanueva-Arce, Garín-Aguilar, Leal-Lara and Valencia-del Toro [44] found that *P. ostreatus* reached a cooked MGC of only 2% for both *P. ostreatus* flour and protein extract. However, it was not possible to determine if this concentration was presented on a wet or dry basis or if evaporation occurred during cooking that changed the effective concentration during gelation. Another study was conducted on cooked and uncooked *A. auricula* and reported an MGC of 8% and 10% (w/v) for uncooked and cooked samples, respectively [45], which is similar to the results found in this study.

### 3.6. Rheology Behavior

The rheological behavior of the materials was quantified due to its importance in food and 3D printing applications. For instance, the viscosity and yield stress of foods are critical to the experience of chewing and swallowing and also the effectiveness of extrusion during 3D printing [9,69]. All of the fungal materials exhibited shear thinning behavior (Figure 5) with flow index (n) values < 1 and minimal evidence of yield stress (Table 5). This indicates that all samples behaved principally as fluids and were appropriately modeled with the power law relationship. The viscosity at a shear rate of 10 s^−1^ (η_10_) and fitting parameters K and n were generally within the range obtained for *A. auricula* gels formulated with 0.3–0.9% gums (xanthan, *k*-carrageenan, and arabic) reported by Xing, Chitrakar, Hati, Xie, Li, Li, Liu and Mo [9], although AP-15%, OP-15%, OF-15%, and OF-25% all displayed low performance as measured by their low η_10_ and K values.

A positive relationship was observed between flour and protein extract inclusion rates and consistency coefficients (K) while a negative relationship generally existed between inclusion rate and flow index (n). These results are consistent with prior investigations with fungal and plant materials and can be described by the formation of intermolecular aggregate structure that is easily disrupted with application of shear stress [9,47]. Decreased values of n indicate enhanced shear-thinning behavior. However, a balance must be struck since excessive values of K require the application of high shear stresses that can make extrusion 3D printing difficult or cause damage to bioink contents (e.g., cells) [9,69,70].

The protein extracts of *A. awamori* demonstrated lower consistency coefficients (K) than their corresponding flours, while the opposite relationship (higher K values) was observed for the *P. ostreatus* protein extracts compared to their flours (Figure 5, Table 5). Both OP-15% and OP-25% protein extracts displayed heightened shear-thinning (lower n) behavior compared to their flours while no consistent relationship was determined between the *A. awamori* protein extracts and flours. These results indicate that the rheological behavior of fungal protein extracts is species-dependent. The larger content of disulfide bonds in *P. ostreatus* protein extract samples compared to *A. awamori* protein extract (Figure 1) could be a contributing factor to the observed rheological difference between material fractions of the two species since disulfide bonds are important contributors to gel strength and stability [71,72,73]. Other interactions from aforementioned polysaccharides, pigments, and other materials in the fungal flour of *A. awamori* played clear roles in its high consistency (K) compared to its protein extract, indicating relatively weaker protein interactions in the materials from this species compared to *P. ostreatus*. It is also important to note that the two protein extracts differed slightly in the protein content of the 15 and 25% inclusion materials (Table A1) with the *A. awamori* protein extracts containing more protein due to their higher protein recoveries. Both protein extracts displayed similar K and n values at low inclusion (15%) with larger observed differences observed at the higher inclusion rate (25%). Overall, the 25%-inclusion protein extract samples had n values of 0.194 and 0.289 for *A. awamori* and *P. ostreatus*, respectively, which were among the best shear-thinning properties observed overall. In the context of food products, the shear-thinning properties of the AP-25% and OP-25% protein extracts were similar to animal egg whites and plant-based egg white analogs (n = 0.28–0.33) [47], while the consistency index values for the OP-25% and AP-25% samples were on the order of salad dressing (K = 15 Pa∙s^n^) [47] and low-inclusion (5–10%) pea, soy, and/or faba bean protein extracts (K = 68–300 Pa∙s^n^) [46], respectively.

The *A. auricula-judae* flour also demonstrated high K and low n rheological performance, which is probably mainly owed to polysaccharide interactions (principal monosaccharides are glucose, mannose, and galactose), an effect that has been documented before [34,74]. Previous investigations observed enhanced shear-thinning behavior (n = 0.176) with *A. auricula-judae* polysaccharide extracts compared to the present results with unextracted flour. When considered together with the difficulties encountered in printing the *A. auricula-judae* flour (discussed in subsequent section), it is recommended that future investigations consider printing polysaccharide extracts of this organism.

### 3.7. Three-Dimensional Bioprinting of Fungal Gels

Due to the innate gelling nature of the fungal samples in this study, the printing of the gels was investigated as one example of an applied use for fungal hydrogels (Figure 6). The pressure required to induce a steady flow during the 3D printing of each sample (Table 1) corresponded closely with the trend of viscosity and consistency indices (Figure 5 and Table 5). Samples with very low K values (i.e., AP-15%, OP-15%, OF-15%, and OF-25%) required low extrusion pressure and could not create stable printed structures. On the other hand, all AF samples were sufficiently viscous (K = 281–694 Pa∙s^n^) but demonstrated less shear-thinning behavior (n = 0.379–0.416) relative to more successfully printed materials such as AP-25% and OP-25% (K = 26–228 Pa∙s^n^, n = 0.194–0.289). These observations are also consistent with the MCG test results, which indicated that passing the MCG test did not necessarily indicate successful printing of homogeneous, smooth, and stable structures. Most notably, the *A. awamori* protein extract showed mixed MGC results at 15% and successful gelation at 20% (Figure 4), but the material did not retain its structure following printing, resulting in puddled material that did not form the shape of the six-pointed star. The other protein extract sample, *P. ostreatus*, showed a similar outcome, but retained more of the structure with the 20% inclusion sample. The resolution of the star shape was satisfactory but collapsed over time. The *A. auricula-judae* flour samples lacked homogeneity and were not able to print successfully. *A. awamori* flour samples also resulted in sputtering during printing, possibly owing to their relatively high K and n values which may have hindered printing resolution and print stability. On the other hand, the protein extract gels appeared highly homogeneous and resulted in stable prints. Lastly, a common issue occurring during printing was drying. The *A. awamori* protein extract and flour mixtures were found to be particularly susceptible to drying in the needle between prints, causing the needle to clog and the print to fail. In general, there appeared to be a positive correlation between inclusion rate and drying susceptibility. This effect is a known challenge of 3D bioprinting and is deserving of future investigation to enable scale-up.

At a 25% inclusion rate, every sample except *A. awamori* flour printed the structure successfully with the best resolution being obtained with *P. ostreatus* protein extract. The resolution of the successful gels varied, with the corners and edges of the two protein extract samples and 25% *P. ostreatus* flour appearing straight and relatively sharp without the appearance of pooling. Besides these three gels, the other samples exhibited pooling without the formation of sharp edges. There is still much work to be performed to improve the materials for the desired applications, such as modifying the color and ensuring better homogeneity, but the protein extract samples exhibited promising characteristics as printable hydrogels.

There has been significant work investigating the functional properties of fungi, but this is the first work to report on the use of fungal flours and protein extracts as the singular ingredient in 3D printing hydrogel studies. The ultimate amount of protein in each hydrogel sample can be shown in Table A1. Most notably, the amount of protein content is similar between the protein extract gels and the flour gels, respectively, of *A. awamori* and *P. ostreatus*. The protein content of the *A. auricula-judae* flour gel was only 0.10 g at 25% inclusion, indicating that protein gelation played a very small role in the gelation of this fungal material. While the amount of protein in *A. awamori* and *P. ostreatus* was similar, the work shown with SDS-PAGE indicated that the two crude protein extract materials contain different specific proteins and potentially slightly different gelation mechanisms.

### 3.8. Texture Profile Analysis of Printed Constructs

Texture Profile Analysis (TPA) was performed to measure the textural characteristics of the printed fungal hydrogels (Figure 7). Only successfully printed fungal hydrogels were subjected to TPA as liquid samples are not appropriate for TPA. Hardness measures gel strength under compressive force (Figure 6A). It was clear that increases in inclusion rate were concordant with increases in the hardness. The *A. awamori* flour had the highest hardness at 41.2 g at 25% inclusion rate, with its protein extract counterpart having a hardness of 25.9 g at the same inclusion amount. Cohesiveness measures the ability a material has to hold together under force. This is important for a gel to have in order to hold together, creating a strong structure. Figure 6B shows that *A. awamori* flour has the highest cohesiveness which increased with inclusion rate. The same was not found for the protein extract samples, with the 20% and 25% both having nearly identical cohesiveness. Gumminess measures the degree of hardness and cohesion for semisolid food products, like hydrogels (Figure 6C). This measure encompasses the results of both hardness and cohesiveness, thus reflecting the same outcome with *A. awamori* flour greatly exceeding the other materials. Resilience is the ability for the gel to reform to its original state after deformation (Figure 6D). It is important for a gel to be resilient to change for certain applications, such as when being used as a scaffolding material in cellular agriculture. Most notably, the *A. awamori* flour displayed higher resilience but not by as significant of a margin as before, with *A. awamori* flour performing very similarly to its protein extract counterpart at 25% with 23.2% and 22.7% resilience, respectively.

Comparing the texture profile results to other studies is difficult when it comes to 3D-printed products. Currently, there is no standard protocol for testing the TPA of a hydrogel or bioink post-printing, which is more reflective of final product characteristics than pre-printed samples. Factors like shape, size, infill amount, temperature, pressure, nozzle type, and more can change the TPA even with the same printing material [75]. This was demonstrated by Kamlow et al. [76] with textural differences in a printed cylinder and square, and casted cylinder and square. The cast samples had a significantly higher hardness compared to the printed samples, whereas the shape of the print did not impact the hardness. More work needs to be carried out in optimizing the textural properties of the printed samples for different targeted applications.

## 4. Conclusions

The fungal materials considered in this work are promising materials for hydrogel formulation. They were rich in protein, and that protein was easily extracted using isoelectric point precipitation to acceptable purity. The protein extracts of *A. awamori* and *P. ostreatus* were found to contain characteristics like disulfide bonds that contribute to protein gelation. FT-IR analysis confirmed the reduction in C-H, N–H, and C–O functional groups in protein extract samples, all of which are groups found in the structure of chitin, the cell wall material of fungi. The flour samples had on average a larger particle size than the protein extracts, but all had overall negative zeta potential. The fungal materials were able to form a gel that was 3D printable with only water, pH adjustments, and heating steps. Formulations of 15–25% protein extract and flour demonstrated shear-thinning characteristics required for 3D printing. Despite substantial textural differences, all the mixtures were able to form gels at 20% inclusion rates. The best performing sample for 3D printing was the *P. ostreatus* protein extract at 25% inclusion. Other samples like the *A. awamori* protein extract and *P. ostreatus* flour also printed acceptably at 25%, but did not perform as well in texture profile (TPA) measurements. *P. ostreatus* flour performed the overall worst in all categories for desired characteristics of alternative protein products, and *A. awamori* protein extract was less hard and gummy compared to *P. ostreatus* protein extract. *A. awamori* protein extract at 25% was far more resilient than other samples and produced higher quality prints than the *A. awamori* flour.

This work illustrates a potential use for fungi as a future food additive. Studies covering fungal flours and protein extracts as food ingredients are rare in the literature and there is a need to investigate other functional properties of these materials considering the wide diversity of fungi species available for sustainable food applications. It is recommended that future work consider the optimization of process parameters (ionic strength, pH, cell wall disruption, etc.) during extraction and gelation and to test different additive materials (e.g., sodium-alginate, transglutaminase) which may improve the strength or reduce the inclusion rate of the gels. Other factors like color were not optimal for food applications with the samples utilized in this study, so bleaching of the materials and observing effects on gelling behavior and printing performance is needed. Overall, this study points towards promising applications of fungal protein extracts as alternative protein food ingredients.

## Figures and Tables

**Figure 1 foods-14-00923-f001:**
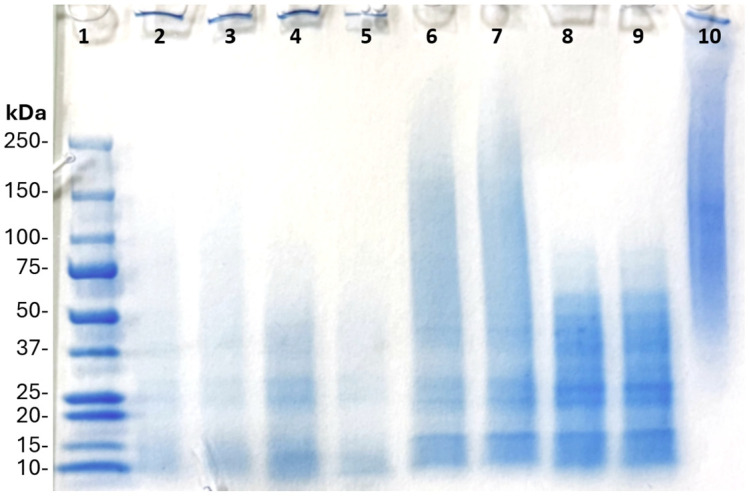
Colorized SDS-PAGE results for A. awamori and P. ostreatus protein extracts. Well labels are: (1) Pre-stained ladder standard, (2) AP, (3) AP+E, (4) AP+RA, (5) AP+E+RA, (6) OP, (7) OP+E, (8) OP+RA, (9) OP+E+RA, (10) gelatin. AP = *A. awamori* protein extract, OP = *P. ostreatus* (oyster mushroom) protein extract, RA = β-mercaptoethanol, E = protease inhibitor cocktail.

**Figure 2 foods-14-00923-f002:**
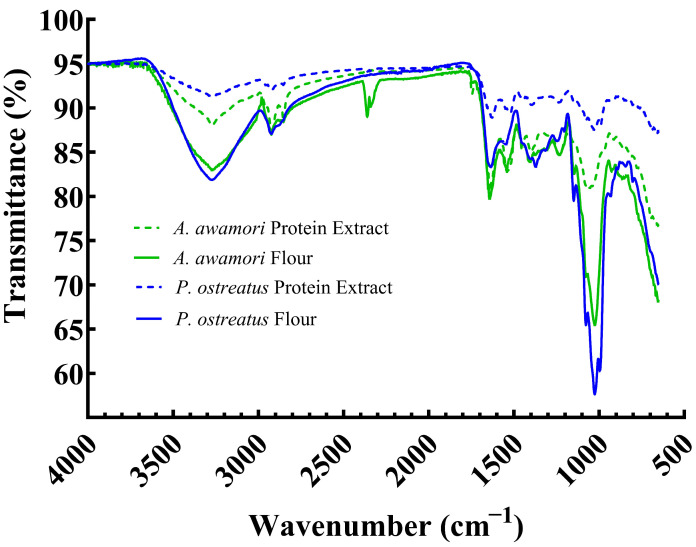
Normalized FT-IR transmittance for *A. awamori* and *P. ostreatus* protein extracts and flours. Demonstrates 4 peaks at 3300, 2900, 1650, and 1000 cm^−1^. Spectra are averages of *n* = 3.

**Figure 3 foods-14-00923-f003:**
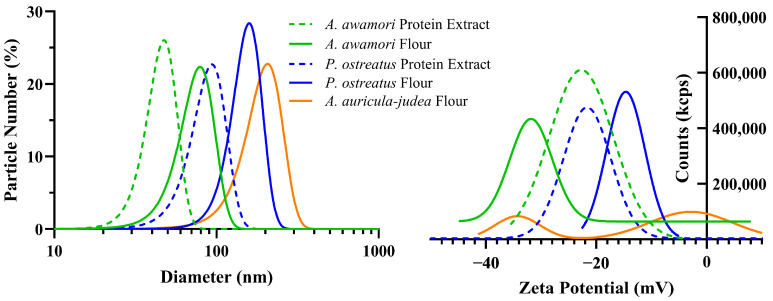
Gaussian distribution of particle size (**left**) and zeta potential (**right**) at pH of 7 for protein extracts and flours.

**Figure 4 foods-14-00923-f004:**
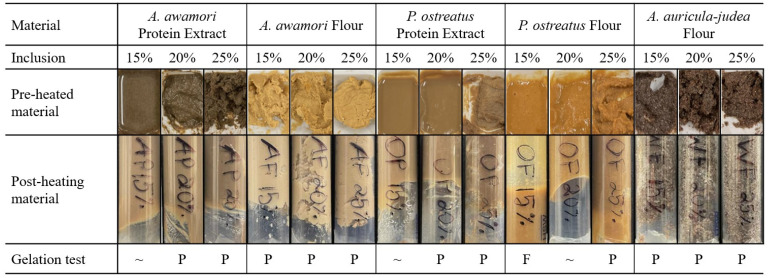
Pre-heated and post-heated inverted cooked mixtures to demonstrate MGC with inclusion rates of 15%, 20%, and 25% (wet basis) and gelation test. F, failure; ~, mixed results; P, passes.

**Figure 5 foods-14-00923-f005:**
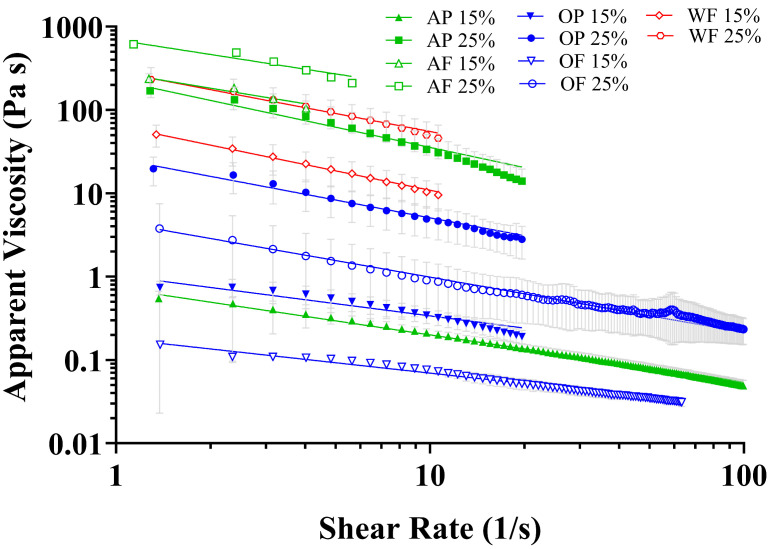
Shear-thinning rheological properties of the fungal flour and protein extract samples. Data are reported as the mean and standard deviation of n = 2 with regression trendlines. AF (*A. awamori* flour), AP (*A. awamori* protein extract), OF (*P. ostreatus*, oyster mushroom, flour), OP (*P. ostreatus*, oyster mushroom, protein extract), WF (*A. auricula-judae*, wood ear mushroom, flour).

**Figure 6 foods-14-00923-f006:**
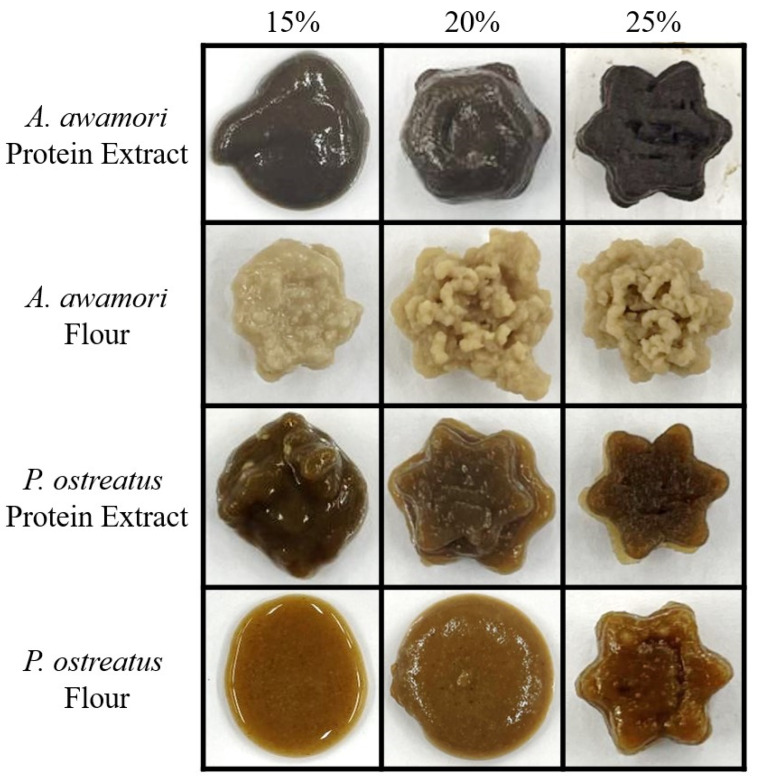
Printed cooked hydrogels using 19-gauge tip with specific speed and pressure settings depending on the material (Table 1).

**Figure 7 foods-14-00923-f007:**
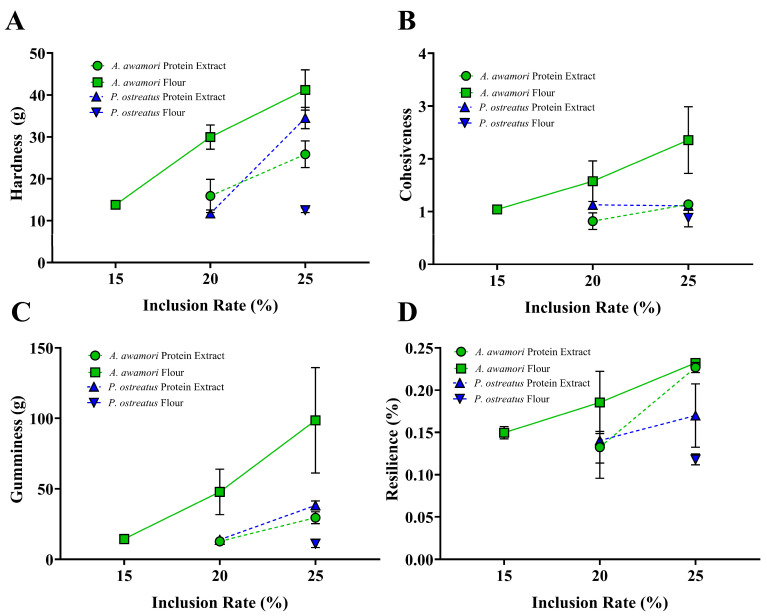
TPA results for successful fungal hydrogel prints with (**A**) hardness, (**B**) cohesiveness, (**C**) gumminess, and (**D**) resilience. Data are reported as the mean and standard deviation of *n* = 2, with *P. ostreatus* protein extract 25% *n* = 3.

**Table 1 foods-14-00923-t001:** Three-Dimensional printing parameters, average pressure and printing speed used for each material.

Material	Inclusion	Pressure (kPa)	Printing Speed (mm/s)
*A. awamori* Protein Extract (AP)	15%	X *	X
	20%	10.3	10
	25%	41.4	4
*A. awamori* Flour (AF)	15%	13.8	6
	20%	55.2	4
	25%	138.0	5
*P. ostreatus* Protein Extract (OP)	15%	X	X
	20%	10.3	8
	25%	41.4	4
*P. ostreatus* Flour (OF)	15%	X	X
	20%	X	X
	25%	10.3	9

* X indicates that the material had an extrusion pressure under 1 psi (unable to be attained with the equipment) and could not be printed into stable structures.

**Table 2 foods-14-00923-t002:** Nitrogen, carbon, and protein contents (dry basis), tapped bulk density, extract yields (g extract/g dry mass), and protein recovery (protein content in the extract divided by the protein content in the fungal flour) from alkaline extraction–isoelectric precipitation. Measurements of 40-mesh screened powders. Data are presented as mean ± standard deviation for applicable samples.

Sample		% Carbon	% Nitrogen	% Protein	Extract Yield (g/g Dry Biomass)	Protein Recovery (%)	Bulk Density (g/mL)
*A. awamori*	Protein extract	50.8 ± 0.1	9.4 ± 0.1	59.0 ± 0.4	0.16 ± 0.0	32.2 ± 0.0	0.45
	Flour	45.3 ± 0.2	4.7 ± 0.0	29.3 ± 0.3			0.20
*P. ostreatus*	Protein extract	43.9 ± 0.0	8.2 ± 0.0	51.5 ± 0.2	0.09 ± 0.0	18.4 ± 0.0	0.61
	Flour	39.4 ± 0.1	4.1 ± 0.1	26.0 ± 0.4			0.43
*A. auricula-judae*	Flour	38.9 ± 0.11	1.0 ± 0.1	6.0 ± 0.9	-	-	0.90

**Table 3 foods-14-00923-t003:** Summarized mean ± standard deviation of particle size and zeta potential values for fungal flour and protein extracts. *A. auricula-judae* zeta potential shows two values representing the observed bimodal distribution.

Sample		Particle Size Diameter (nm)	Zeta Potential (mV)
*A. awamori*	Flour	79.22 ± 18.62	−31.61 ± 5.07
	Protein Extract	47.48 ± 9.85	−22.71 ± 5.91
*P. ostreatus*	Flour	159.1 ± 32.55	−14.65 ± 3.49
	Protein Extract	93.67 ± 21.82	−21.65 ± 4.31
*A. auricula-judae*	Flour	206.6 ± 50.95	−34.27 ± 4.12 −2.838 ± 7.543

**Table 4 foods-14-00923-t004:** Color parameters of pre-heated, hydrated fungal flour and protein extract materials at different inclusion levels. Values are presented as mean ± standard deviation (n = 3).

Sample	Inclusion (%)	L*	a*	b*
*Aspergillus awamori* protein extract	15	38.2 ± 0.6	4.5 ± 0.0	18.9 ± 0.1
	20	37.6 ± 1.6	3.7 ± 0.1	18.8 ± 0.4
	25	23.6 ± 0.8	4.2 ± 0.3	17.7 ± 0.6
*Aspergillus awamori* flour	15	65.8 ± 0.8	8.7 ± 0.3	42.4 ± 0.4
	20	61.3 ± 1.5	9.7 ± 0.7	43.6 ± 0.4
	25	65.9 ± 0.2	8.3 ± 0.2	42.8 ± 1.0
*P. ostreatus* protein extract	15	48.8 ± 0.5	7.8 ± 0.1	30.6 ± 0.3
	20	49.0 ± 0.2	7.7 ± 0.1	30.0 ± 0.2
	25	49.3 ± 2.4	7.9 ± 0.7	25.7 ± 1.4
*P. ostreatus* flour	15	55.0 ± 0.6	17.1 ± 0.1	46.6 ± 0.6
	20	56.1 ± 0.5	16.5 ± 0.4	44.7 ± 0.6
	25	52.4 ± 0.4	16.5 ± 0.9	41.4 ± 1.7
*A. auricula-judae* flour	15	31.4 ± 0.2	5.7 ± 0.1	10.4 ± 0.1
	20	26.4 ± 1.4	6.7 ± 0.4	12.9 ± 1.0
	25	31.9 ± 1.3	7.9 ± 0.2	13.5 ± 0.3

**Table 5 foods-14-00923-t005:** Viscosities at shear rate 10 s^−1^ (η_10_) and power law model fitting parameters for fungal flour and protein extract samples. AF (*A. awamori* flour), AP (*A. awamori* protein extract), OF (*P. ostreatus*, oyster mushroom, flour), OP (*P. ostreatus*, oyster mushroom, protein extract), WF (*A. auricula-judae*, wood ear mushroom, flour).

Formulation	Power Law			
	η_10_ (Pa∙s)	K (Pa∙s^n^)	n	R^2^
AP-15%	0.20	0.73	0.439	0.955
AP-25%	35.56	227.60	0.194	0.770
AF-15%	67.21	280.90	0.379	0.884
AF-25%	180.85	694.40	0.416	0.896
OP-15%	0.34	1.04	0.513	0.785
OP-25%	5.08	26.12	0.289	0.776
OF-15%	0.07	0.18	0.585	0.952
OF-25%	0.99	4.53	0.338	0.580
WF-15%	10.88	64.69	0.226	0.797
WF-25%	55.18	288.40	0.282	0.738

## Data Availability

The data presented in this study are available on request from the corresponding author due to the policy of the funding agency and authors’ institutions.

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
