# Peer review of "Evaluation of the Gelation Characteristics and Printability of Edible Filamentous Fungi Flours and Protein Extracts"

_foods, 2025, doi:10.3390/foods14060923_

Round 1
Reviewer 1 Report
Comments and Suggestions for Authors
Evaluation of the Gelation Characteristics and Printability of Edible Filamentous Fungi Flours and Protein Extracts
This study presents an innovative approach by exploring new ingredients, specifically proteins from alternative sources, for the development of 3D-printed foods using filamentous fungi. It is a pioneering study that provides valuable insights for future research.
Suggestions and Recommendations
- Objective and Contextualization: The authors provide a good introduction to the use of fungal protein, its future prospects, and the need for further studies. However, it would be relevant to include, in the introduction, the objective of obtaining 3D structures from fungal material and highlighting its industrial applicability, particularly in the production of 3D protein gels. The 3D printing technology could also be further explored in terms of practical applications.
- Color Characterization: It is recommended that the authors present instrumental color measurements of the samples, using, for example, the CIELAB system. The way the results were discussed makes the explanation of color changes vague. The statement "The observed color change is likely due to a chemical reaction occurring with the pigments in fungal materials." is too generic and lacks a more in-depth scientific discussion, with examples and explanations based on the literature.
- Consistency in Numerical Data: For greater clarity, it is suggested to standardize the values presented in both the text and tables, avoiding inconsistent rounding. For example, in the statement "The A. auricula-judea flour displayed the highest bulk density of 0.897 g/mL while the A. awamori flour was lowest at 0.195 g/mL," the values should be consistently maintained between different formats of presentation.
- Organization of Discussion and Methodology: The discussion section contains passages that are more appropriate for the methodology, such as the following paragraph:
"The protein contraction rate was calculated by creating a ratio of the percent protein of the protein extract content over the percent protein of the flour for each fungus. The purity of the protein was evaluated by observing the final protein content of the crude protein extracts."
This passage should be moved to the methodology section to ensure a clearer structure for the article. - Missing or Poorly Detailed Data: Some information needs to be better presented, such as the following points:
- The data mentioned in "Overall, the total protein extraction efficiency was 32.24% and 18.42% for A. awamori and P. ostreatus, respectively." are not properly located in the text. Where are these results presented?
- How was the carbon percentage determined? This procedure is not clearly described.
- Discussion Based on Literature: The discussion could be strengthened with more references from the literature, especially regarding techniques such as ATR-FTIR, Zeta Potential, and Particle Size for the characterization of flours and protein extracts.
- Color Characterization and Analytical Methodology: As previously mentioned, the authors state that "The color of the A. awamori protein was found to significantly darken compared to the flour, changing from a light tan to a dark brown color." However, why was an instrumental analytical system not used for this characterization? The use of a quantitative color scale would enhance the analysis.
- Determination of Minimum Gelation Concentration (MGC): The method used to observe MGC was empirical ("The minimum gelation concentration (MGC) was observed by inverting the heated gels and noting if the materials adhered to the sides or slid down."). Why was a more appropriate methodology not used, such as tests based on the flow properties of viscous materials (e.g., inclined plane, spreadability, or another standardized method)? This would add greater scientific rigor to the results.
Author Response
Comments and Suggestions for Authors
Evaluation of the Gelation Characteristics and Printability of Edible Filamentous Fungi Flours and Protein Extracts
This study presents an innovative approach by exploring new ingredients, specifically proteins from alternative sources, for the development of 3D-printed foods using filamentous fungi. It is a pioneering study that provides valuable insights for future research.
The authors extend our gratitude to the reviewer for appreciating the work and for providing thoughtful comments and suggestions, which have been comprehensively addressed in our revision.
Suggestions and Recommendations
- Objective and Contextualization: The authors provide a good introduction to the use of fungal protein, its future prospects, and the need for further studies. However, it would be relevant to include, in the introduction, the objective of obtaining 3D structures from fungal material and highlighting its industrial applicability, particularly in the production of 3D protein gels. The 3D printing technology could also be further explored in terms of practical applications.
- We have added significant discussion of the background and practical applications of 3D printing as well as more specific examples utilizing fungi in lines 44-63.
- Color Characterization: It is recommended that the authors present instrumental color measurements of the samples, using, for example, the CIELAB system. The way the results were discussed makes the explanation of color changes vague. The statement "The observed color change is likely due to a chemical reaction occurring with the pigments in fungal materials." is too generic and lacks a more in-depth scientific discussion, with examples and explanations based on the literature.
- We agree that this inclusion is more scientific and objective. We analyzed our hydrated flour and protein extracts with an image processing software (ImageJ) and have added relevant methods and discussion to lines 194-201 and 403-414, respectively. We have included an additional table with the L*A*B* parameters (Table 4).
- Consistency in Numerical Data: For greater clarity, it is suggested to standardize the values presented in both the text and tables, avoiding inconsistent rounding. For example, in the statement "The A. auricula-judea flour displayed the highest bulk density of 0.897 g/mL while the A. awamori flour was lowest at 0.195 g/mL," the values should be consistently maintained between different formats of presentation.
- Thank you. We have made the modification in this section and have checked the manuscript and made appropriate changes throughout as needed.
- Organization of Discussion and Methodology: The discussion section contains passages that are more appropriate for the methodology, such as the following paragraph:
"The protein contraction rate was calculated by creating a ratio of the percent protein of the protein extract content over the percent protein of the flour for each fungus. The purity of the protein was evaluated by observing the final protein content of the crude protein extracts."This passage should be moved to the methodology section to ensure a clearer structure for the article. - We agree and have added this methodology to lines 153-157, which now clearly states the calculations utilized for protein yield, recovery, and purity.
- Missing or Poorly Detailed Data: Some information needs to be better presented, such as the following points:
- The data mentioned in "Overall, the total protein extraction efficiency was 32.24% and 18.42% for A. awamori and P. ostreatus, respectively." are not properly located in the text. Where are these results presented?
- We agree that this was not clear before. In addition to the methodology additions (see previous comment), we also significantly revised and reorganized the first subsection of Section 3 (Results and Discussion) to make this clearer.
- How was the carbon percentage determined? This procedure is not clearly described.
- Thank you for noticing this error. We utilized the Dumas combustion method for but forgot to include carbon in the methods section. It has been added to line 150.
- Discussion Based on Literature: The discussion could be strengthened with more references from the literature, especially regarding techniques such as ATR-FTIR, Zeta Potential, and Particle Size for the characterization of flours and protein extracts.
- We have strengthened the discussion and included more references in these sections.
- Color Characterization and Analytical Methodology: As previously mentioned, the authors state that "The color of the A. awamori protein was found to significantly darken compared to the flour, changing from a light tan to a dark brown color."? The use of a quantitative color scale would enhance the analysis.
- As noted above, we have updated this to include an objective measurement of L*A*B* for the flours and protein extracts.
- Determination of Minimum Gelation Concentration (MGC): The method used to observe MGC was empirical ("The minimum gelation concentration (MGC) was observed by inverting the heated gels and noting if the materials adhered to the sides or slid down."). Why was a more appropriate methodology not used, such as tests based on the flow properties of viscous materials (e.g., inclined plane, spreadability, or another standardized method)? This would add greater scientific rigor to the results.
- We agree that the MGC is not an ideal analysis but it is widely used for characterizing flours and protein extracts and importantly, we noticed that the MGC concentrations found in this study were higher than previously reported for filamentous fungi. To obtain quantitative flow behavior and supplement the MGC result, we have included rheological analysis.
Reviewer 2 Report
Comments and Suggestions for Authors
The research is of interest in terms of protein extraction of mushroom material and the research adds to the current pool of information on this area. The manuscript is organised well and the writing is mostly suitable.
Please avoid the generic terms of comparison such as "good" when evaluating the quality (such as rheological properties) of the mixtures. Be very specific.
The extraction techniques are sound, however the composition of the material reveal the protein extract to be about 50% of the weight of the material? Is this correct.
If this is correct, can you please provide a more comprehensive analysis of the ither significant components present in the material as these will affect the rheological and functional behavior of the product.
I would suggest that the authors should discuss the presence of different polysaccharides in the materials (the rheological changes between groups is most likely significantly affected by differences in polysaccharide components and the gelling and rheological water binging properties).
Can you give a table of fibre composition and also profiles ? Possibly also water and oil holding capacities in relation to polysaccharide composition.
There needs to be a greater discussionof this aspect in the paper.
Author Response
Comments and Suggestions for Authors
The research is of interest in terms of protein extraction of mushroom material and the research adds to the current pool of information on this area. The manuscript is organised well and the writing is mostly suitable.
Thank you very much for your feedback and thoughtful comments, which have helped us improve the quality of this work.
Please avoid the generic terms of comparison such as "good" when evaluating the quality (such as rheological properties) of the mixtures. Be very specific.
Thank you for this suggestion. We have modified the rheology and printing sections to remove this subjective language and strengthen the objectiveness of the discussion.
The extraction techniques are sound, however the composition of the material reveal the protein extract to be about 50% of the weight of the material? Is this correct.
Yes, the total protein recovery was low although the purity ranged from 51.1 – 59.0% in the extracts. We have enhanced the discussion of this calculation and included specific methodology to lines 153-157.
If this is correct, can you please provide a more comprehensive analysis of the ither significant components present in the material as these will affect the rheological and functional behavior of the product.
More discussion of other components of fungi including polysaccharides has been included to the FTIR section (lines 341-348 and 357 - 369) as well as the color discussion (lines 395 – 406). Important polysaccharide interactions for A. auricula-judea are included in the rheological discussion of this material in lines 460 - 467.
I would suggest that the authors should discuss the presence of different polysaccharides in the materials (the rheological changes between groups is most likely significantly affected by differences in polysaccharide components and the gelling and rheological water binging properties).
As indicated above, we have substantially increased discussion of the nature of polysaccharides that may be present in the samples under study.
Can you give a table of fibre composition and also profiles ? Possibly also water and oil holding capacities in relation to polysaccharide composition.
Fiber composition of related materials has been included in the ATR-FTIR section (line 343 – 369). Unfortunately, we could not present comprehensive profiles or impacts on other properties (water or oil holding capacity) due to a lack of primary data in the present study. We have indicated areas of future work and hope you agree that the present study is a suitable first step towards those future inquiries.
There needs to be a greater discussionof this aspect in the paper.
Thank you. We believe we have sufficiently addressed your comments and have substantially increased the depth of analysis and discussion in this work as a result.